# Scale-aware Recognition in Satellite Images under Resource Constraints

**Shreelekha Revankar**[1]*, **Cheng Perng Phoo**[1], **Utkarsh Mall**[2], **Bharath Hariharan**[1], **Kavita Bala**[1]

[1]Cornell University [2]Columbia University

## Abstract

Recognition of features in satellite imagery (forests, swimming pools, etc.) depends strongly on the spatial scale of the concept and therefore the resolution of the images. This poses two challenges: Which resolution is best suited for recognizing a given concept, and where and when should the costlier higher-resolution (HR) imagery be acquired? We present a novel scheme to address these challenges by introducing three components: (1) A technique to distill knowledge from models trained on HR imagery to recognition models that operate on imagery of lower resolution (LR), (2) a sampling strategy for HR imagery based on model disagreement, and (3) an LLM-based approach for inferring concept "scale". With these components we present a system to efficiently perform scale-aware recognition in satellite imagery, improving accuracy over single-scale inference while following budget constraints. **Our novel approach offers up to a 26.3% improvement over entirely HR baselines, using 76.3% fewer HR images.** Resources are available on our website.

Figure 1: With these images we can see how concept scale is linked to spatial resolution. If we are seeking out a spatially large concept like *forest*, lower resolutions are favored *(b)*, as higher resolutions may lack the needed context to discern between a forest *(a)* and a park *(c)*. At the same time while seeking out finer concepts such as *sports track*, certain details can only be discerned well at higher resolutions *(d)* and are obscured at lower resolutions *(e)*.

## 1 Introduction

The ever-increasing number of earth observation satellites (now more than 1500) offer us a unique opportunity to understand changes at the planetary scale, be it tracking the ecological degradation of forests and coral reefs (Wicaksono et al., 2021; Gao et al., 2020), the destruction of cultural heritage (Tapete et al., 2021), or economic development and well-being (Engstrom et al., 2022). A key to all these downstream applications is the ability to recognize accurately a broad vocabulary of concepts: a major computer vision challenge.

An important aspect of recognition in satellite imagery is the notion of *scale*; satellite image resolution is dependent on the particular satellite/sensor and is characterized by the GSD, or ground

---

*Corresponding Email: revankar@cs.cornell.edu

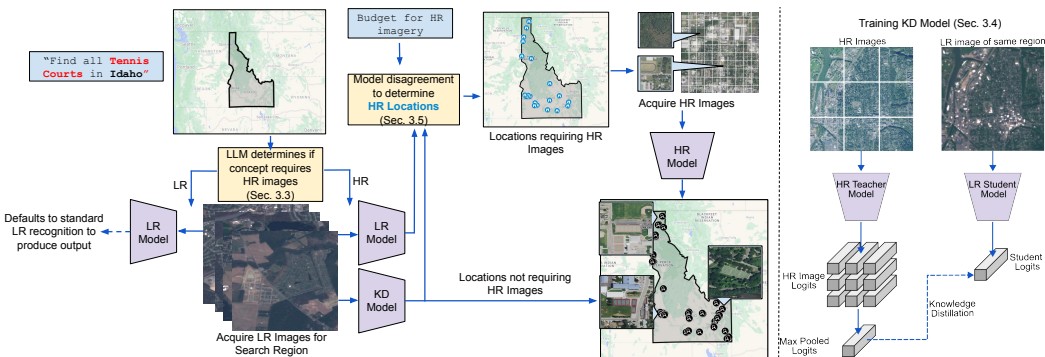

Figure 2: System overview. First, we determine which resolution is best suited for the search concept based on its scale (Sec. 3.3). Then, we analyze the search area to find which regions would benefit the most from higher resolution inference (Sec. 3.5). We sample the best suited regions while staying within a user specified budget. Based on this guidance we perform inference using one of three models, a high resolution satellite model, a low resolution satellite model, and a low resolution satellite model with knowledge distilled from its high resolution counterpart (Sec. 3.4). This knowledge distilled model allows us to infer finer details using low resolution satellite imagery alone.

sampling distance: the distance corresponding to a single pixel. For example, Sentinel-2 images are fairly low resolution (1 pixel = 100 $m^2$) (ESA, 2024), while images captured by the NAIP program are fairly high resolution (1 pixel = 1 $m^2$) (U.S. Geological Survey, 2024). The image scale matters because many concepts of interest also have a characteristic physical size/scale. An olympic-sized swimming pool, typically about 50 m long, would be barely a speck in Sentinel-2 images. In contrast, a lake may be hundreds of square kilometers, necessitating an exceptionally large image to capture its extent in terms of NAIP imagery, which current models would not be equipped to do.

The right resolution may also depend on the underlying geography. Dense urban areas with many geographical features may require high resolution data to analyze. In contrast, large swathes of uninhabited deserts can probably be analyzed accurately even with low-resolution imagery.

Accuracy notwithstanding, we must also consider cost and availability. Low-resolution (LR) imagery is free, abundant, and covers the planet densely over space and time (ESA, 2024; USGS Earth Resources Observation and Science, 2024). In contrast, high resolution (HR) imagery typically comes from drones or low flying satellites that are commissioned as needed, and therefore come with a high cost and inconsistent temporal and spatial coverage (Planet Labs PBC, 2018–; U.S. Geological Survey, 2024).

Scientists in many application domains, be it ecology, archaeology, and urban planning (Wicaksono et al., 2021; Gao et al., 2020; Tapete et al., 2021; Engstrom et al., 2022), today manually reason about these different aspects, namely, the scale of the concept, the nature of the underlying geography, and the cost of acquired imagery, to achieve the best recognition that they can afford with their budget. We need an approach that can automate this tradeoff and automatically identify when to acquire more expensive high resolution imagery, taking into account the cost, the scale of the concept and the geography. While there is past work on the impact of scale on satellite image recognition, much of it is focused on accuracy and disregards costs (Reed et al., 2023; Mall et al., 2024; Bastani et al., 2023). Prior work that has looked at costs focuses on the geography but ignores the scale of the concept itself Uzkent & Ermon (2020). A holistic treatment of scale to achieve accurate recognition under a budget is missing.

In this paper, we address this gap by proposing:

- A knowledge distillation technique which allows LR models to improve significantly in recognizing finer concepts by learning from HR.
- A novel approach that leverages the semantic understanding of LLMs to determine the scale of each concept.

- A new approach that determines which geographical regions require higher resolution analysis by predicting when low and high resolution models might disagree.
- A unified framework that combines these ideas to yield the most accurate retrieval results for a range of concepts while adhering to strict budget constraints.

Finally, we note that while our experiments are on LR and HR satellite imagery, these techniques are more general and can generalize to any problem where there are two or more modalities with different costs and accuracy tradeoffs.

We evaluate our approach with multiple recognition models (supervised and open-vocabulary) and multiple satellite modalities. We find that compared to simply using high resolution images always, our framework *improves* accuracy up to 13 points while reducing the number of HR images used by 5×. Our approach also significantly outperforms (by more than 25 points) other prior work that trades off between accuracy and cost. In sum, our results demonstrate that our holistic reasoning of scale leads to significantly higher accuracy with large reductions in cost.

## 2    RELATED WORKS

### 2.1    MULTI-SCALE RECOGNITION IN SATELLITE IMAGERY

Remotely sensed images or satellite images are inherently multi-scale. Depending on the sensor heights and properties, the physical distance between two adjacent pixels in remotely sensed images (known as Ground Sample Distance/GSD) could vary from 0.3m to 1km. These multi-scale images provide complementary information for various applications (Wicaksono et al., 2021; Gao et al., 2020; Tapete et al., 2021; Engstrom et al., 2022).

Few prior works have explicitly investigated the effect of different scales in visual recognition in satellite imagery. For instance, Reed et al. (2023) shows that it is crucial to build scale-specific representations for images with different GSD. However, they do not explicitly address the high acquisition cost of high-resolution satellite imagery. Works on image super-resolution (Shermeyer & Van Etten, 2019; Kowaleczko et al., 2022; Wolters et al., 2023) seek to sidestep the problem of acquisition cost by upsampling low-resolution to their high-resolution counterparts. But as shown in Wolters et al. (2023), these models are prone to hallucination and would produce high-resolution imagery with limited fidelity. In this work, we consider a more realistic setup where we allow a small budget for acquiring high-resolution images during inference. This allows us to tap into the stronger performance of scale-aware models such as Reed et al. (2023); Mall et al. (2024); Bastani et al. (2023) while avoiding the hallucination problem in image super-resolution.

### 2.2    RESOURCE CONSTRAINTS IN VISUAL RECOGNITION

Various resource constraints could be considered during the development or deployment of visual recognition models. Active learning approaches (Tuia et al., 2011) consider a budget for acquiring annotations during development and seek to develop the best-performing models by selecting the most informative set of training examples to annotate. This could be done by selecting examples that the model is most uncertain about (Houlsby et al., 2011; Shannon, 2001; Yoo & Kweon, 2019), examples that are most representative of the unlabeled data (Ash et al., 2019; Sener & Savarese, 2017), or a combination of both (Yin et al., 2017). Prior works have also considered compute constraints for developing and deploying visual recognition models. These works satisfy the compute constraints either through developing specialized model architectures (Tan & Le, 2019) or compressing/pruning existing models (Hinton et al., 2015; Blalock et al., 2020). Our work considers a different type of constraint inherent to remote sensing, i.e., the acquisition cost for high-resolution images. While others have looked into optimizing high-resolution image acquisition (Uzkent & Ermon, 2020; Meng et al., 2022), we provide a solution to produce the performant visual recognition models under a fixed acquisition budget with a semantic understanding of the concept. To the best of our knowledge, this is an uncharted problem domain.

## 2.3 KNOWLEDGE DISTILLATION/LEARNING WITH PRIVILEGED INFORMATION

To recognize fine-grained concepts in LR images, we construct a query model by distilling the logits generated by another model that is trained on HR imagery. This approach bears a resemblance to knowledge distillation (Hinton et al., 2015; Borup & Andersen, 2021; Cho & Hariharan, 2019; Park et al., 2019; Fukuda et al., 2017; Borup et al., 2023), which seeks to compress a large-scale teacher model by training a compute-efficient student model to mimic the output of the teacher model. Different from conventional knowledge distillation, our query student model and teacher model operate on different inputs. Specifically, our query model operates on LR imagery, whereas our teacher model operates on HR imagery. Our approach can also be viewed as an instance of learning with privileged information (Vapnik & Vashist, 2009; Pechyony & Vapnik, 2010; Vapnik et al., 2015), in which additional information that is available during test time (e.g., high-resolution satellite imagery) is used for training a model. This paradigm has enabled the development of better-performing models in various applications, such as autonomous driving (Chen et al., 2019), image super-resolution (Lee et al., 2020), and so on. In remote sensing, prior works (Kumar et al., 2021; Li et al., 2022) have attempted to use additional sensor data as privileged information for improving land-cover classification. Our work leverages high-resolution satellite image data as privileged information for improving recognition in low-resolution imagery: a different scenario.

## 3 METHODOLOGY

### 3.1 PROBLEM SETUP

We are interested in defining a framework for optimizing accuracy/cost tradeoffs when performing recognition in satellite imagery. This framework should reason about not only the scale of the concept being recognized but also the underlying geography of each location.

We concretize the problem as follows.

**Setup.** We frame our problem as a *retrieval task*: given a concept $c$ (e.g., "tennis courts") and a large set of locations $\mathcal{I}$ (e.g., the state of New York), we wish to identify which of these locations have the concept.

For each location, we have one LR image and a set of HR images that tile the same area. The size of this set, $K$, depends on the difference in GSD between the two modalities. For example, if LR images come from Sentinel-2 (GSD=10m) and HR images come from NAIP (GSD=1m), $K$ would be 100.

We assume that we have special purpose multi-label classification models for each resolution, denoted by $M^{HR}$ and $M^{LR}$ respectively. Given any concept $c$ and any location $i$, we can use these models to score the presence of this concept at that location. For LR imagery, this is straightforward:

$$s_c^{LR}(i) = [M^{LR}(I_i^{LR})]_c \tag{1}$$

For HR imagery, we simply take the maximum over model scores for all the images at that location:

$$s_c^{HR}(i) = \max_{j=1}^{K} [M^{HR}(I_{ij}^{HR})]_c \tag{2}$$

The max here is used because we want $s_c^{HR}(i)$ to be high (denoting the presence of the concept) if *any* of the $K$ HR images indicate the presence of the concept. We call these scores LR scores and HR scores respectively.

As discussed in the introduction, HR imagery is expensive. We assume a budget in terms of the maximum number of HR images we can acquire, $B$. Our goal is, given a concept $c$ and a set of locations $\mathcal{I}$, retrieve all locations $i \in \mathcal{I}$ that have concept $c$ while staying within the HR budget.

**Training / hyperparameter validation.** To allow for algorithms to train any models or validate hyperparameters, we assume access to a fixed set of locations $\mathcal{I}^{train}$ where we have already acquired (a) images from both resolutions, and (b) annotations for the presence/absence of a set of "seen" concepts $\mathcal{C}^{seen}$. For fairness, these locations are distinct from the locations used for evaluation. Our set of evaluation concepts are also kept distinct ("unseen" classes); however, we also report performance on the seen classes. A full list of the concepts, seen and unseen, is in Appendix A.1.

## 3.2 SYSTEM OVERVIEW

Our overall system to tackle this problem is shown in Figure 2 and has three steps: Given a search concept, our system uses an LLM-based approach to predict which resolution it is best suited towards (Sec. 3.4). If the search concept was found to be better suited to HR imagery, we would ask the user for a budget and utilize our Model Disagreement based sampling technique to optimize the budget in selecting locations for HR imagery (Sec. 3.5). We describe how this technique can be performed using LR imagery alone via a Knowledge-Distilled LR model and how such a model allows us to perform improved inference with LR imagery alone as well (Sec. 3.4).

## 3.3 INFERRING CONCEPT "SCALE" USING LLMS

The question of whether HR imagery is needed or not depends on the concept. For some concepts, the low-resolution scores $s^{LR}$ are better, while for others, the HR scores $s^{HR}$ are more accurate. This is, unfortunately, difficult to predict without some background knowledge about the concept. To capture this background knowledge, we propose to leverage large language models (LLMs).

Unfortunately, while LLMs have generally learned various kinds of background knowledge, it is unlikely that they have seen or reasoned about the different modalities of satellite imagery. As such, while the LLM might know that a lake is bigger than a swimming pool, this may not be enough to figure out which of the two modalities is better.

We address this knowledge gap with *in-context learning* (Dong et al., 2022). Concretely, we first use the available training locations $\mathcal{I}^{train}$ and the annotations therein to evaluate both the HR scores and the LR scores for each concept in $\mathcal{C}^{seen}$. Thus, for each of these seen concepts, we know which modality yields higher accuracy. We then give these seen concepts, as well as the corresponding modality, as in-context examples to the LLM and ask the LLM to infer the right modality for all other (unseen) concepts. The prompts to the LLM can be found in Sec. A.2 in the appendix.

## 3.4 KNOWLEDGE DISTILLATION FROM HR TO LR

Suppose we know apriori that a concept is better detected in HR imagery. Given the limited acquisition budget of $B$, we do not have full coverage of the locations in $\mathcal{I}$. Thus we need a model that consumes LR images and approximates the predictions made by $M^{HR}$. To build such a model $M_{KD}^{LR}$, we minimize the MSE between the model's prediction and the corresponding HR scores:

$$\sum_{i} \sum_{c \in \mathcal{C}^{seen}} ||[M_{KD}^{LR}(I_i^{LR})]_c - s_c^{HR}(i)||_2^2 \tag{3}$$

By minimizing the loss, we essentially distill the HR knowledge (Hinton et al., 2015) in the $M^{HR}$ into a model that consumes LR images $M_{KD}^{LR}$. This formulation also bears a resemblance to learning with privileged information (Vapnik et al., 2015). In this case, when training the model $M_{KD}^{LR}$, we use privileged HR information to guide the $M_{KD}^{LR}$ to detect smaller concepts at LR imagery.

## 3.5 ACQUIRING HR IMAGERY BASED ON MODEL DISAGREEMENT

When detecting a small concept, $M_{KD}^{LR}$ might not be perfect since an LR image at the test location $l$, $I_l$ might not have enough visual signatures. In this case, we would have to leverage our budget of $B$ to acquire HR imagery and run predictions using $M^{HR}$. A simple criterion to select a location $l$ to acquire HR imagery is by looking at the disagreement between the LR scores and HR scores. The disagreement at $l$ can thus be defined as:

$$\delta(l) = \sum_{c \in \mathcal{C}^{seen}} |s_c^{HR}(l) - s_c^{LR}(l)| \tag{4}$$

A location $l$ with higher $\delta(l)$ is more likely to have concepts that $M^{HR}$ could detect where $M^{LR}$ could not (or vice versa). However, to compute $\delta$ we would have to already possess HR imagery over the whole region, which is exactly what we want to avoid acquiring. Instead to approximate

$\delta$, we leverage $M_{KD}^{LR}$ as a stand in for $M^{HR}$ and compute the following disagreement criterion for ranking various locations:

$$\delta'(l) = \sum_{c \in \mathcal{C}^{seen}} |[M_{KD}^{LR}(I_l^{LR})]_c - s_c^{LR}(l)| \tag{5}$$

### 3.6 SCALE-AWARE RECOGNITION UNDER RESOURCE CONSTRAINTS

Given a concept our LLM determines which resolution is best suited (Sec. 3.3). If the best suited resolution is LR, we perform inference using the LR model and all LR imagery only. If the best suited resolution is HR, we ask the user for their HR budget. Then we perform model disagreement using LR imagery alone and the LR and KD models using eq. 5. With these disagreement scores and the user budget, we then sample the top locations, obtain HR images and perform inference using the HR model for these locations. The retrieval results for the remaining area, that it is preferable to use HR for, but cannot be fit in the budget, is scored using the KD model on LR imagery. The end result is accurate retrieval where HR imagery is used sparingly, only when it is really needed.

## 4 EXPERIMENTS AND RESULTS

We evaluate the effectiveness of our scale-aware concept recognition approach by assessing the impact of knowledge distillation, model-disagreement-based acquisition, and LLM-based scale inference. We first evaluate our whole system then the three components individually.

### 4.1 EXPERIMENTAL SETUP

#### 4.1.1 PRE-TRAINED MODELS

We perform our experiments with two sets of low-resolution and high-resolution models.
- GRAFT is a *zero-shot* vision-language model that can be used for performing open-vocabulary recognition on satellite imagery (Mall et al., 2024), similar to CLIP (Radford et al., 2021). We use the released model for Sentinel-2 and NAIP and train our own model for the NICFI basemaps. The model architecture is ViT-B-16.
- We finetune a *fully-supervised* ResNet-50 pre-trained on ImageNet on both LR (Sentinel-2) and HR (NAIP) data with Open Street Maps annotations on our training set.

#### 4.1.2 BENCHMARK

For our experiments, we require aligned LR and HR imagery to fairly compare the performance of LR and HR models. However, this is not required in our problem setup or approach. We curate the following two benchmarks. Both benchmarks use Sentinel-2 as the low-resolution modality, but differ in the high-resolution modality:

**Sentinel-2/NAIP.** In this benchmark, we use HR imagery (GSD=1m) captured by the National Agriculture Imagery Program (U.S. Geological Survey, 2024). This program only captures data for the US, so this benchmark is restricted to the US. Here, one Sentinel-2 image corresponds to 100 NAIP images. The cost of acquiring similar resolution imagery ranges from \$1.00–\$6.00 per square km (Geocento, 2023).

We created a training dataset and validation dataset using images from the following states and regions: Arkansas, Delaware, Idaho, Maine, Rhode Island, Wyoming, and US Virgin Islands. These datasets are comprised of 45,885 Sentinel-2 images for LR, and 4,588,500 NAIP images for HR in the training dataset, and 4,938 and 493,800 images respectively for the validation dataset. Our testing imagery is comprised of images from D.C., Puerto Rico, and Hawaii. The test dataset is comprised of 5,015 Sentinel-2 and 505,100 NAIP images. We chose these locations as they represent a diverse set of demographic and climatic regions within the US.

**Sentinel-2/NICFI.** Our second benchmark leverages the NICFI Satellite Data Program Basemaps for Tropical Forest Monitoring (Planet Team, 2024) which provides HR imagery (GSD=5m) for

regions of Central/South America, Africa, and Asia. Similar imagery has prices ranging from $0.60–$2.0 per square km (Geocento, 2023). Here one Sentinel-2 image corresponds to 4 NICFI images.

All data was sourced from Google Earth Engine (Gorelick et al., 2017). To minimize temporal discrepancies, LR images were collected within the same month as their HR counterparts. Following Mall et al. (2024) we use OpenStreetMap contributors (2024) to obtain ground truth annotations for 40 concepts (listed in Appendix A.1). All data and download scripts will be publicly released.

All classes are *seen* for supervised techniques, while all classes are *unseen* for zero-shot baselines. We reserve 30 of these classes as *seen* classes and the other 10 are *unseen* classes when evaluating the knowledge-distilled zero-shot techniques.

### 4.1.3 METRICS

Since the scope of this method is retrieval, we evaluate our results with **mAP**@k averaged over all classes. We also look at the amount of HR imagery utilized, as well as runtime in seconds to perform inference on the imagery. The runtime is calculated on the images determined by each technique; for LR models our test set has 5,015 images, and for our HR models the test set has 501,500 images. Techniques utilizing HR sampling methods, such as our own or Patchdrop (Uzkent & Ermon, 2020) use a combination of these two datasets. All inference is performed using a batch size of 32 on a single Nvidia RTX A6000 GPU. For our experiments we set a budget of 1000 locations to acquire HR imagery, with each location covering roughly 5 sq. km. We report our results in sq. km., so we can compare to other techniques that do not sample by location.

### 4.2 RESULTS

We examine our method by testing individual components. We first evaluate the entire system (Sec. 4.2.1). We then evaluate our LR Knowledge Distillation Models in Sec. 4.2.2. After that, we examine our model disagreement scheme for selecting modalities in Sec. 4.2.3). Finally, we assess the usage of LLMs for search concept scale determination (Sec. 4.2.4).

### 4.2.1 OVERALL SYSTEM PERFORMANCE

We assess the performance of our entire system (Table 2) by comparing the following baselines:

- **HR**: Simply using the HR data and HR models for all locations.
- **LR**: Only using LR data and LR model.
- **KD**: Only using LR data, but with our KD model.
- **model dis.**: Using our model disagreement approach to sample locations for HR imagery, but without any LLM-based inference of whether the concept needs HR data.
- **LR + LLM**: Using an LLM to decide which concepts need HR data, but then acquiring HR images for all locations and using the LR model for other concepts.
- **KD + LLM**: Using an LLM to decide which concepts need HR data, but then acquiring HR images for all locations and using the KD model for other concepts.
- **Patchdrop**: Using Patchdrop (Uzkent & Ermon, 2020) to determine HR sample locations.
- **Ours full**: Using our full system, from LLM to determine the best suited resolution, with our model disagreement technique to sample HR imagery with a constrained budget, and using our KD model to perform inference for HR-suited concepts in out-of-budget area. Which would be the same as KD + LLM + model dis.
- **nl. sampling**: Using sampling strategies put forth in IS-count (Meng et al., 2022) (via-night lights) and compare it to our own model disagreement technique.

We find that our overall system performs the best on both classes of pre-trained models, even outperforming the baseline of using all HR data for all locations. With 26.3% and 24.6% improvement on zero-shot technique GRAFT, and 6% and 8.3% improvement on supervised techniques in mAP[100] and mAP[20] respectively **using only 23.6% of the HR images**.

We see our system perform more efficiently with regard to runtime as our system can handle retrievals for multiple concepts at once. If all of the concepts are best suited for LR, inference using the LR model occurs at most once. If some concepts are better suited for HR imagery then inference

| Model | Data | Seen Classes* | | Unseen Classes | | # HR images | Inf. Time |
|---|---|---|---|---|---|---|---|
| | | $mAP^{100}$ | $mAP^{20}$ | $mAP^{100}$ | $mAP^{20}$ | sq. km. | s |
| GRAFT HR | HR | 0.501 | 0.513 | 0.541 | **0.574** | 25,163 | 1539 |
| GRAFT LR + LLM[†] | LR,HR | 0.530 | 0.561 | **0.549** | 0.573 | 25,163 | 1575 |
| GRAFT KD + LLM[†] | LR,HR | 0.557 | 0.576 | 0.541 | **0.574** | 25,163 | 1570 |
| GRAFT LR | LR | 0.482 | 0.507 | 0.379 | 0.471 | 0 | 17 |
| GRAFT KD[†] | LR | 0.534 | 0.559 | 0.490 | 0.503 | 0 | 17 |
| RemoteCLIP Liu et al. (2024) | LR | 0.461 | 0.493 | 0.413 | 0.440 | 0 | 17 |
| CLIP-RSICD Arutiunian et al. (2021) | LR | 0.393 | 0.441 | 0.311 | 0.303 | 0 | 17 |
| OpenCLIP Cherti et al. (2023) | LR | 0.481 | 0.505 | 0.392 | 0.447 | 0 | 16 |
| GRAFT LR + nl. sampling Meng et al. (2022) | LR,HR | 0.379 | 0.354 | 0.440 | 0.492 | **5,954** | 323 |
| GRAFT KD + model dis.[†] | LR,HR | 0.621 | 0.628 | 0.461 | 0.494 | **5,954** | 340 |
| **Graft (Ours full)**[†] | LR,HR | **0.633** | **0.639** | 0.502 | 0.564 | **5,954** | 372 |

Table 1: Full system performance under various settings for retrieval under budget for zero-shot techniques.[†]:Techniques utilizing our contributions. *: Classes *seen* during knowledge distillation, for all other zero-shot baselines **all** classes are *unseen*. For models using both HR and LR data, we assign a budget of 1000 locations ($\sim$ 5 sq. kms each). Our full system performs much better than the baselines with improved HR data constraints.

| Model | Data | $mAP^{100}$ | $mAP^{20}$ | # HR images in sq. km. | Inf. Time in s |
|---|---|---|---|---|---|
| Supervised HR | HR | 0.695 | 0.735 | 25,163 | 1585 |
| Supervised LR + LLM[†] | LR,HR | 0.689 | 0.731 | 25,163 | 1610 |
| Supervised KD + LLM[†] | LR,HR | 0.696 | 0.735 | 25,163 | 1609 |
| Supervised LR | LR | 0.451 | 0.473 | 0 | 14 |
| Supervised KD[†] | LR | 0.570 | 0.606 | 0 | 14 |
| Resnet 50 (Multi-Res) | LR | 0.421 | 0.447 | 0 | 21 |
| SatMAE Cong et al. (2022) | LR | 0.351 | 0.388 | 0 | 67 |
| Cross-Scale MAE Tang et al. (2023) | LR | 0.412 | 0.422 | 0 | 91 |
| Supervised LR + nl. sampling Meng et al. (2022) | LR,HR | 0.463 | 0.480 | **5,954** | 319 |
| Supervised + Patchdrop Uzkent & Ermon (2020) | LR,HR | 0.445 | 0.470 | 6,776 | 360 |
| Supervised KD + model dis.[†] | LR,HR | 0.733 | 0.791 | **5,954** | 328 |
| **Supervised (Ours full)**[†] | LR,HR | **0.736** | **0.796** | **5,954** | 338 |

Table 2: Full system performance of various fully-supervised techniques for retrieval under budget.[†]:Techniques utilizing our contributions. For models using both HR and LR data, we assign a budget of 1000 locations ($\sim$ 5 sq. kms each). Our full system performs much better than the baselines with improved HR data budget constraints.

happens at most twice using LR imagery and at most once using HR imagery. So irrespective of the number of concepts, we can retrieve multiple concepts in about the same time.

### 4.2.2 RECOGNITION IN LOW RESOLUTION

We compare the performance of several models for multi-label classification in Tables 3 and 4 on 40 classes (as listed in the appendix A.1) using LR imagery alone. We see improvements in performance through our knowledge distillation process for both zero-shot and fully-supervised techniques. For each corresponding training data type and model architecture, we also show the performance of the HR model (in gray).

We also compare our zero-shot KD Model with other zero-shot baselines CLIP-RSICD (Arutiunian et al., 2021) and RemoteCLIP (Liu et al., 2024) in Table 3. We compare our fully supervised KD Models with other fully supervised baselines ScaleMAE (Reed et al., 2023), SatMAE (Cong et al., 2022), and Cross-Scale MAE (Tang et al., 2023) as well as a Resnet-50 fine-tuned on imagery of multiple resolutions (Multi-Res) in Table 4.

While the performance of our KD models reaches closer to that of HR models there still is a gap (e.g., more than 5 percentage points in mAP). This suggests that knowledge distillation from HR data alone is not enough to build better recognition systems. Therefore, a framework like ours that works on multiple resolutions is needed, since not all concepts are favored by HR imagery.

### 4.2.3 MODEL DISAGREEMENT FOR MODALITY SELECTION

Following the methodology in Sec. 3.5, we describe how the disagreement between HR and LR models can act as a good indicator for which locations would benefit from HR imagery. We found that correlation coefficient between the disagreement scores of the HR (NAIP) vs KD with the LR model is **0.9322**, which signifies a strong positive correlation. Similarly, the correlation coefficient

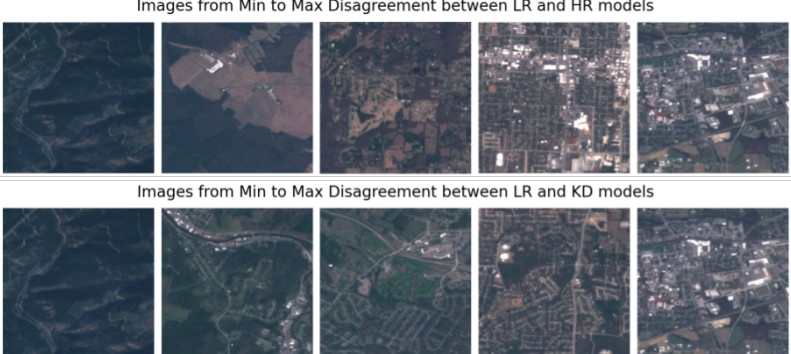

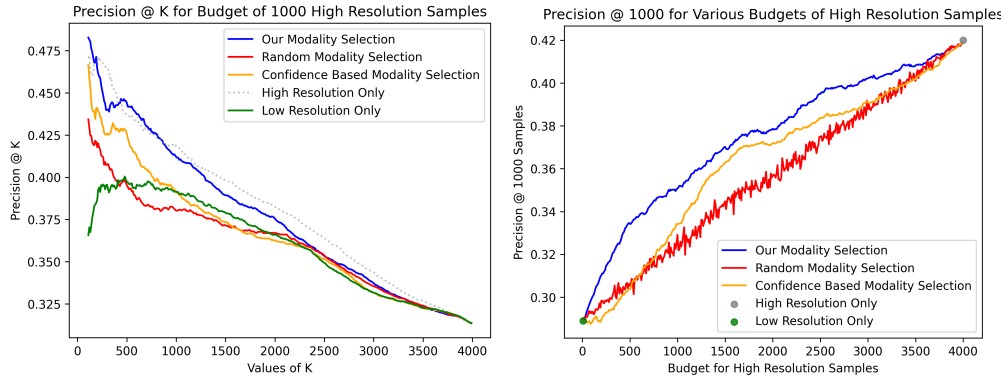

Figure 3: Images ranked according to disagreement between the LR and HR model (top) and the LR and KD model (bottom). Both rankings are similar, with a correlation coefficient of **0.9322**, even though the latter only uses LR images.

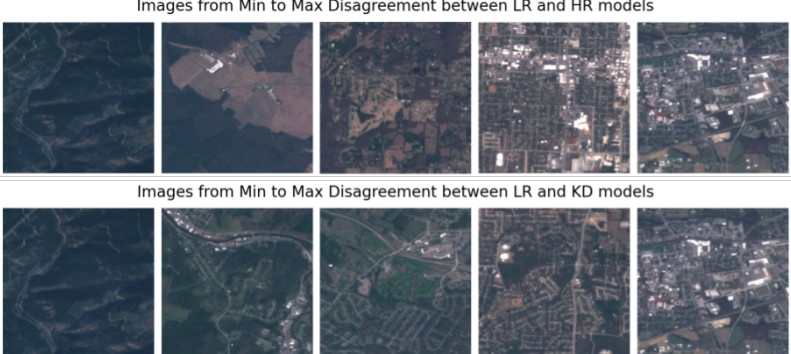

Figure 4: Performance when using our model disagreement-based sampling strategy. Our approach consistently yields higher precision across all budgets and across all values of $K$.

between the disagreement scores of the HR (NICFI) vs KD with the LR model is **0.998**, which also signifies a strong positive correlation.

Figure 3 visualizes images ranked by degree of disagreement between the LR and HR model (top) and between the LR and KD model (bottom). We see that both approaches yield similar rankings demonstrating that the KD model can act a stand in for the HR model, so that we may calculate the disagreement without requiring HR imagery.

We use this model disagreement along with the KD and HR model to show how well we perform retrieval under an HR-data budget. Figures 4 illustrate the improvement in retrieval with our HR data sampling technique. The figure on the left shows precision@K at a fixed HR data budget (1000 locations) and the figure on the right shows precision@1000 when varying the data budget for the GRAFT model on unseen categories. Our HR data sampling strategy is significantly better than randomly sampling locations or model uncertainty-based methods. This also shows that sampling HR images leads to improvement over just using LR-data with KD models.

We compare the performance of our sampling technique against the sampling technique presented in (Meng et al., 2022), in which night lights are used as an indicator to sample more densely. The results of this sampling technique are included in Table 2.

### 4.2.4 CONCEPT SCALE INFERENCE

We compare our LLM-based approach for inferring the best modality per concept with other baseline approaches. For each approach, we evaluate its accuracy in terms of determining the correct modality. These baselines include either always choosing the HR modality and always choosing LR modality, or selecting between the two based on the average area of each concept, as provided via

| Model | Seen Classes* | | Unseen Classes | |
|---|---|---|---|---|
| | $mAP^{100}$ | $mAP^{20}$ | $mAP^{100}$ | $mAP^{20}$ |
| Sentinel-2/NAIP | | | | |
| GRAFT HR | 0.501 | 0.513 | 0.541 | 0.574 |
| GRAFT LR | 0.482 | 0.507 | 0.379 | 0.471 |
| RemoteCLIP | 0.461 | 0.493 | 0.413 | 0.440 |
| CLIP-RSICD | 0.393 | 0.441 | 0.311 | 0.303 |
| OpenCLIP | 0.481 | 0.505 | 0.392 | 0.447 |
| **GRAFT KD** | **0.534** | **0.559** | **0.490** | **0.503** |
| Sentinel-2/NICFI | | | | |
| GRAFT HR | 0.338 | 0.412 | 0.424 | 0.473 |
| GRAFT LR | 0.202 | 0.204 | 0.311 | 0.339 |
| **GRAFT KD** | **0.300** | **0.369** | **0.414** | **0.425** |

Table 3: Performance of the unsupervised models for recognition at low resolution. Our knowledge-distilled models perform better on both seen and unseen concepts (corresponding HR models in grey; references included in Tab. 1).

| Model | $mAP^{100}$ | $mAP^{20}$ |
|---|---|---|
| Sentinel-2/NAIP | | |
| Supervised HR | 0.695 | 0.735 |
| Supervised (Multi-Res) | 0.421 | 0.447 |
| Scale-MAE | 0.472 | 0.493 |
| SatMAE | 0.351 | 0.388 |
| Cross-Scale MAE | 0.412 | 0.422 |
| Supervised LR | 0.451 | 0.473 |
| **Supervised KD** | **0.570** | **0.606** |
| Sentinel-2/NICFI | | |
| Supervised HR | 0.699 | 0.850 |
| Supervised LR | 0.659 | 0.801 |
| **Supervised KD** | **0.681** | **0.837** |

Table 4: Performance of the supervised models for concept recognition at low resolution. Our knowledge-distilled models perform better over all classes of concepts (corresponding HR models in grey; references included in Tab. 2).

OpenStreetMaps (OSM). Our LLM approach (tested using several off-the-shelf models) achieves 100% accuracy when determining the correct modality our set of concepts. We test on both *seen* and *unseen*, demonstrating the ability of easily being extended to far more concepts.

Both the OSM average area approach and choosing only HR imagery were correct 90% of the time whereas selecting solely LR imagery was correct only 10% of the time. This result is interesting as the LLM outperformed using the average area of the concepts (OSM), suggesting that it is not just the size of the concept that is an important consideration, but also a semantic understanding of the concept and its features. We include a tabulated version of these results in the appendix A.3.

**Additional Experiments.** We include the following additional experiments in the Appendices to further exhibit the effectiveness of our approach. We demonstrate the impact of different budgets, ranging from 100 locations to 1000 in A.4. Additionally, we compare our KD model with the HR model on Super-Resolution images to calculate model disagreement scores in A.5.

**Limitations.** While our technique works well to cover all concepts regardless of scale, it performs in a setting in which satellite imagery is split into two groups, low and high resolution. However, in reality, resolution is not discrete; recognizing exactly which resolution is best suited to a concept is something we have not yet determined.

We also utilize OSM data to train and evaluate our models. We acknowledge that this data contains noise, due to its crowd-sourced nature. Unfortunately, there are not many sources for fine-grained labeling of satellite imagery. Therefore, we follow past work in using OSM for training and evaluation (Mall et al., 2024; Bastani et al., 2023). Moreover other standard sources of labeled satellite imagery such as Microsoft's building footprint dataset work with similar noise levels (Bing Maps, 2018). Additionally, to show robustness to this noise, we performed an experiment using the GRAFT LR model, wherein we constructed bootstrap samples by sub-sampling within our original test set 1000 times. The results show that the 90% confidence interval is within 2.43% of the reported mAP.

Our technique also does not allow for the segmentation of concepts, i.e., segmenting specific regions within lower-resolution images that are worth looking into at a higher resolution. We leave the exploration of this direction to future work.

## 5 CONCLUSION

**Conclusions.** We introduce a new approach to scale-aware recognition in satellite imagery under resource constraints. Our approach allows one to accurately detect various concepts using a fixed budget of HR imagery, outperforming entirely HR baselines by more than 26% mAP in zero-shot techniques, and more than 8% mAP in supervised techniques using $5\times$ fewer HR images.

**Broader Impacts.** Our system offers a cost-efficient way to perform recognition in satellite imagery by maximizing HR imagery usage while cutting costs. This can have a significant positive impact on a wide range of scientists, anthropologists, archaeologists, NGOs, and human rights organizations, who operate on limited budgets, but greatly benefit by the use of satellite imagery in their work.

**Acknowledgments** We would like to acknowledge Chia-Hsiang Kao for his help in revising the abstract. This work was partly funded by the NSF IIS 2403015, NSF IIS 2403016, and USDA NIFA 2023-67021-39829.

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
