# A Appendix

## A.1 Multi-Label Classes

The classes we use for labeling our data as well as for testing are the following.

'tennis', 'skate', 'american football', 'swimming', 'cemetery', 'garage', 'golf', 'roundabout', 'parking lot', 'supermarket', 'school', 'marina', 'baseball', 'fall', 'pond', 'airport', 'beach', 'bridge', 'religious', 'residential', 'warehouse', 'office', 'farmland', 'university', 'forest', 'lake', 'nature reserve', 'park', 'sand', 'soccer', 'equestrian', 'shooting', 'ice rink', 'commercial area', 'garden', 'dam', 'railroad', 'highway', 'river', 'wetland'

In the case of the unsupervised techniques, the first 30 are used as the seen classes, while the latter 10 are treated as unseen classes.

## A.2 Prompts to LLM used when Inferring Concept Scale

The prompt was *"Please act as a binary classifier for the following concepts to determine if they are better suited for 'LR' (low resolution) or 'HR' (high resolution) imagery. I will first give you some examples with the correct response. Then given a concept you are to return 'lr' or 'hr'"*

This was followed by the list of 'seen' concepts and their optimal resolution in the format *"concept:resolution"*. After this the LLM only responded with either *'lr'* or *'hr'* when given a concept.

## A.3 Tabulated Concept Scale Inference Results

| Technique | Seen Classes Accuracy | Unseen Classes Accuracy |
|---|---|---|
| LLM ChatGPT 3.5 (ours) | **100%** | **100%** |
| LLM Gemini (ours) | **100%** | **100%** |
| LL Claude (ours) | **100%** | **100%** |
| OSM | 73.33% | 90% |
| Always HR | 83.33% | 90% |
| Always LR | 16.67% | 10% |

Table 5: Comparison of concept recognition performance (accuracy, precision, recall) between baseline models and LLM selected models.

We compared our LLM-based approach for inferring the best modality per concept with other baseline approaches. For each approach, we evaluate its accuracy in terms of determining the right modality. These baselines include either always choosing the HR modality and always choosing LR modality, or selecting between the two based on the average area of each concept, as provided via OpenStreetMaps (OSM) OpenStreetMap contributors (2024).

We evaluate on the 10 unseen classes as the *seen* concepts were given to the LLM as examples as described in Sec. A.2.

The LLM approach selected the correct resolution 100% of the time. Both the OSM average area approach and choosing only HR imagery were correct 90%. Finding that the LLM outperformed using the average area of the concepts (OSM), suggests that it is not just the size of the concept that is an important consideration, but also a semantic understanding of the concept itself.

## A.4 Impact of Different Budgets

We are able to get an improved accuracy on a tight budget since we only sample HR images when necessary. The reason we are able to do so is (a) some concepts are better in LR and (b) for the remaining concepts we are able to recognize which regions require HR. Of course if the budget decreases further, at some point, the accuracy will also decrease. We demonstrate this in table 6.

| Model | Data | Seen Classes* | | Unseen Classes | | # HR images | Inf. Time |
|---|---|---|---|---|---|---|---|
| | | $mAP^{100}$ | $mAP^{20}$ | $mAP^{100}$ | $mAP^{20}$ | sq. km. | s |
| GRAFT HR | HR | 0.501 | 0.513 | 0.541 | 0.574 | 25,163 | 1539 |
| GRAFT LR | LR | 0.482 | 0.507 | 0.379 | 0.471 | 0 | 17 |
| GRAFT (Ours full), Budget 100 | LR,HR | 0.557 | 0.605 | 0.439 | 0.521 | **595** | 48 |
| GRAFT (Ours full), Budget 500 | LR,HR | 0.601 | 0.626 | 0.480 | 0.542 | 2,977 | 170 |
| GRAFT (Ours full), Budget 750 | LR,HR | 0.614 | 0.617 | 0.495 | 0.557 | 4,466 | 247 |
| **GRAFT (Ours full), Budget 1000** | LR,HR | **0.633** | **0.639** | 0.502 | 0.564 | **5,954** | 372 |

Table 6: Performance of our overall technique utilizing various budgets. *: Classes *seen* during knowledge distillation, for all other zero-shot baselines **all** classes are *unseen*. We are able to get an improved accuracy on a tight budget since we only sample HR images when necessary.

| Model | Data | Seen Classes* | | Unseen Classes | | # HR images | Inf. Time |
|---|---|---|---|---|---|---|---|
| | | $mAP^{100}$ | $mAP^{20}$ | $mAP^{100}$ | $mAP^{20}$ | sq. km. | s |
| GRAFT HR | HR | 0.501 | 0.513 | 0.541 | 0.574 | 25,163 | 1539 |
| GRAFT LR + SR + LLM[†] | LR,HR | 0.527 | 0.583 | 0.512 | 0.547 | **5,954** | 12,867 |
| GRAFT **KD** + SR + LLM[†] | LR,HR | 0.554 | 0.610 | **0.556** | **0.589** | **5,954** | 12,884 |
| **GRAFT (Ours full)**[†] | LR,HR | **0.633** | **0.639** | 0.502 | 0.564 | **5,954** | **372** |

Table 7: Comparison between KD model for disagreement score calculation and HR model on Super Resolution (SR) imagery. [†]:Techniques utilizing our contributions. *: Classes *seen* during knowledge distillation, for all other zero-shot baselines **all** classes are *unseen*. While SR does offer improvements in mAP for unseen classes, **the inference time alone is 34 times longer than our technique, and this improvement only comes with the use of our KD model**.

## A.5 USE OF SUPER RESOLUTION MODEL FOR MODEL DISAGREEMENT SCORE

Here we present the results using the SR images with the HR model to calculate the model disagreement score to sample HR imagery.

Generating and performing inference using the SR data proved to be significantly more computationally expensive. Here we report inference time, one can see the process is significantly slower than using the KD model on the LR images. This is due to the fact that the SR model accepts 32x32 pixel patches of Sentinel-2 images at a time. So for our test set of 5015 LR images of size 256x256 pixels, the SR model is performing inference on 320,960 patches. This process took 12,549.90 s ( 3.5 hours). Additionally upon stitching these SR patches to recreate the coverage of the original LR image, we must split the stitched SR image into 100 components corresponding to the same organization of HR images. This leads to inference using the HR model on 501,500 images. And even after this, one would still need to run the KD model for regions not covered by the budget. Overall this significant jump in inference time makes using any superresolution model in place of our KD model less favorable.

Additionally, our KD model is specifically trained to identify cases where HR data would provide meaningful improvements in recognition accuracy. In contrast, SR models attempt to reconstruct all high-frequency details, regardless of their importance to the recognition task. This targeted approach makes our method more efficient at identifying truly necessary cases for HR image use.

Here we present the results using the SR images with the HR model to calculate the model disagreement score to sample HR imagery. LR + SR + LLM denotes the use of both the LLM and model disagreement components of our work, (using the SR model disagreement), without the use of any KD model. KD + SR + LLM denotes results with our full system, but with the model disagreement scores being obtained via the SR imagery.

While it does offer improvements in mAP for unseen classes, **the inference time alone is 34 times longer than our technique, and this improvement only comes with the use of our KD model.**

## A.6 RESULTS PER CONCEPT

Table 8 contains results for each class, allowing one to see which models performed best between the supervised HR model and the novel technique presented in this paper. One can see that for most classes, our model manages to outperform the HR model, without needing nearly as many HR images, and for some concepts with no HR images at all.

| | Technique | | | | | |
|---|---|---|---|---|---|---|
| | Supervised HR | | | Supervised Ours Full | | |
| Concept | $mAP^{100}$ | $mAP^{20}$ | # HR imgs | $mAP^{100}$ | $mAP^{20}$ | # HR imgs |
| Tennis | 0.617 | 0.701 | 25,163 | **0.969** | **1.0** | 5,954 |
| Skate Park | **0.277** | 0.300 | 25,163 | 0.163 | **0.345** | 5,954 |
| American Football | **0.449** | **0.663** | 25,163 | 0.265 | 0.305 | 5,954 |
| Swimming | 0.789 | 0.781 | 25,163 | **0.912** | 0.879 | 5,954 |
| Cemetery | 0.294 | 0.308 | 25,163 | **0.615** | **0.817** | **0** |
| Garage | 0.696 | 0.835 | 25,163 | **0.964** | **1.0** | 5,954 |
| Golf | 0.483 | 0.194 | 25,163 | **0.974** | **1.0** | **0** |
| Roundabout | 0.401 | 0.471 | 25,163 | **0.823** | **0.984** | 5,954 |
| Parking Lot | **1.0** | **1.0** | 25,163 | **1.0** | **1.0** | 5,954 |
| Supermarket | **0.928** | **0.978** | 25,163 | 0.773 | 0.756 | 5,954 |
| School | 0.824 | 0.717 | 25,163 | **0.913** | **0.956** | 5,954 |
| Marina | **0.389** | **0.516** | 25,163 | 0.189 | 0.294 | 5,954 |
| Baseball | **0.672** | **0.747** | 25,163 | 0.658 | 0.713 | 5,954 |
| Pond | 0.921 | 0.967 | 25,163 | 0.888 | 0.923 | 5,954 |
| Airport | 0.558 | 0.843 | 25,163 | **0.590** | **0.934** | 5,954 |
| Beach | **0.970** | **0.995** | 25,163 | 0.910 | 0.921 | 5,954 |
| Bridge | **0.986** | **1.0** | 25,163 | 0.976 | **1.0** | 5,954 |
| Religious Buildings | 0.933 | **1.0** | 25,163 | **0.980** | **1.0** | 5,954 |
| Residential | **1.0** | **1.0** | 25,163 | **1.0** | **1.0** | 5,954 |
| Warehouse | 0.901 | **0.969** | 25,163 | **0.945** | 0.927 | 5,954 |
| Office | 0.955 | **0.995** | 25,163 | **0.961** | 0.914 | **0** |
| Farmland | **0.859** | **0.883** | 25,163 | 0.854 | **0.883** | **0** |
| University | 0.505 | 0.448 | 25,163 | **0.652** | **0.920** | **0** |
| Forest | **0.977** | 0.995 | 25,163 | 0.933 | **1.0** | **0** |
| Lake | 0.206 | 0.458 | 25,163 | **0.257** | **0.496** | 5,954 |
| Nature Reserve | **0.938** | **0.997** | 25,163 | 0.915 | 0.995 | 5,954 |
| Park | **1.0** | **1.0** | 25,163 | 0.983 | 0.997 | **0** |
| Sand Pits | 0.847 | 0.811 | 25,163 | **0.983** | **1.0** | 5,954 |
| Soccer | **0.497** | **0.440** | 25,163 | 0.122 | 0.053 | 5,954 |
| Equestrian | **0.395** | **0.497** | 25,163 | 0.083 | 0.200 | 5,954 |
| Shooting Range | **0.01** | **0.05** | 25,163 | **0.01** | **0.05** | 5,954 |
| Commercial Area | 0.483 | 0.425 | 25,163 | **0.895** | **0.934** | 5,954 |
| Garden | 0.815 | 0.839 | 25,163 | **0.901** | **0.956** | **0** |
| Dam | **0.062** | **0.227** | 25,163 | **0.062** | **0.227** | 5,954 |
| Railroad | **0.964** | **0.974** | 25,163 | 0.900 | **0.974** | 5,954 |
| Highway | **1.0** | **1.0** | 25,163 | **1.0** | **1.0** | **0** |
| River | **1.0** | **1.0** | 25,163 | 0.967 | **1.0** | 5,954 |
| Wetland | **0.827** | 0.925 | 25,163 | 0.824 | **0.957** | 5,954 |

Table 8: Performance of various algorithms for various concepts. One can see that our usage of HR images is fully concept aware, and we are able to even outperform models that use solely HR imagery.