# OpenReview forum: "Scale-aware Recognition in Satellite Images under Resource Constraints"
_ICLR.cc/2025/Conference — ICLR 2025 Poster_

### Official Review · Reviewer_CazH · 2024-10-28

**Soundness:** 3
**Presentation:** 3
**Contribution:** 3
**Rating:** 6
**Confidence:** 3

**Summary:**

A scene classification task under resource-constrained conditions is proposed, combining image retrieval and scene classification. Knowledge distillation techniques and an active learning-like sample selection strategy are used to choose HR samples, and an LLM is employed to determine the scale of concepts.

**Strengths:**

The authors address the real-world issue where high-resolution (HR) remote sensing images are paid, while low-resolution (LR) images are free, by proposing a framework that effectively improves scene classification performance under limited funding. This framework can be used to analyze which high-resolution images are more beneficial to purchase, helping researchers make informed decisions,

**Weaknesses:**

1. The description of the LLM section is not very clear. The authors did not specify which LLM was used, how the hyperparameters were set, or provide comparative experiments on the performance of different LLMs. This weakens the contribution of the LLM. It seems that the authors are merely using it as a tool for invocation.

2. In Section 4.2.3, the discussion of Figure 3, which uses only five images to demonstrate ranking similarity, is not convincing. The authors need to provide a quantitative analysis to support this argument.

**Questions:**

1. Does Equation 5 require knowledge of all the information for all seen categories at a specific location to be computed? Why are the HR score and LR score related to location? Does this limit the model to only locations that appeared in the training set? How would the model handle unknown locations?

2. Is using LLM-based in-context learning to determine the scale reliable? Has the accuracy been tested on unseen categories in the validation set? Since the context is determined by the HR and LR scores, why not directly train a network using the HR and LR scores? It is well known that LLMs are very resource-intensive, and using an LLM just to determine the scale seems like overkill. Furthermore, it contradicts the premise of resource efficiency—if one cannot afford HR images, they likely don't have the computational resources to run an LLM either.

---

> ### Author Response · Authors · 2024-11-17
> **Review Response**
>
> Thank you for your review and comments. We hope to address your concerns and answer your questions below:
>
> 1. We use an off-the shelf LLM, specifically ChatGPT 3.5 for inferring scales of concepts. LLMs being utilized as tools to infer semantic information in the field of remote sensing has previously generally been focused on improving the understanding of satellite images. Here, instead we leverage the knowledge of these large models in the correct way by prompting using in-context examples. Details of these examples are described in section 6.3 (in the supplementary material). Here we compare our LLM approach to other baselines, and we will include additional results using other off the shelf LLMs (Gemini, Claude). We are happy to provide more details if you have any further questions.
>
> | Technique | Seen Classes Accuracy | Unseen Classes Accuracy |
> |:-----------:|:---------------------:|:----------------------:|
> | LLM ChatGPT 3.5 (ours) | **100%** | **100%** |
> | LLM Gemini (ours) | **100%** | **100%** |
> | LLM Claude (ours) | **100%** | **100%** |
> | OSM | 73.33% | 90% |
> | Always HR | 83.33% | 90% |
> | Always LR | 16.67% | 10% |
>
> We compared our LLM-based approach for inferring the best modality per concept with other baseline approaches. For each approach, we evaluate its accuracy in terms of determining the right modality. These baselines include either always choosing the HR modality and always choosing LR modality, or selecting between the two based on the average area of each concept, as provided via OpenStreetMaps (OSM). Here one can see that the LLM performs with 100% accuracy for our set of concepts, but it also capable of easily being extended to far more concepts.
>
> 2. Figure 3 visualizes the images with varying levels of disagreement. We agree that including a quantitative analysis would strengthen our results and will revise the paper to include details regarding the correlation between the two sets of model disagreement scores. The correlation coefficient between the disagreement scores of the HR (NAIP) vs KD with the LR model is 0.9322, which signifies a strong positive correlation. The correlation coefficient between the disagreement scores of the HR (NICFI) vs KD with the LR model is 0.998, which also signifies a strong positive correlation.
>
> Questions:
> 1. The models are not limited to locations only seen during training. Eq. 5 pertains to calculating this disagreement between LR and KD for a location. We assume we can acquire LR images for all locations at inference time. We can then calculate this disagreement score using the LR and KD models with the LR image.
>
> 2. We discuss testing the LLM on **unseen** classes in both sec. 4.2.4 on page 9 as well as more in depth on page 2 of the supplementary materials, with Table 5 tabulating these results on our validation set and comparing our technique to others.
>
> The cost of querying an LLM with a few words is essentially free compared to the cost of acquiring HR satellite imagery: one does not need to host a LLM locally, we simply query existing LLMs. We used ChatGPT 3.5 for our experiments, which is free, and we only need to query the LLM once for any given concept, since once we know which resolution is optimal, we don’t need to query again. As for why we do not train a network to estimate the HR and LR scores, we want to ensure we can generalize to unseen concepts.

---

> > ### Comment · Reviewer_CazH · 2024-11-20
> > **The response addressed my concerns.**
> >
> > Thank you for your detailed explanations and clarifications. Your response addressed my previous concerns, and I now understand your work better. I am satisfied with the improvements made and am happy to raise my score from 5 to 6.

---

> > > ### Author Response · Authors · 2024-11-26
> > > **reply to comment**
> > >
> > > Thank you, we are glad we have addressed your concerns and have updated the pdf with results and clarifications from the review discussion.

---

### Official Review · Reviewer_Kyd3 · 2024-11-03

**Soundness:** 3
**Presentation:** 2
**Contribution:** 3
**Rating:** 6
**Confidence:** 5

**Summary:**

This paper addresses the challenge of feature recognition in satellite imagery, focusing on the optimal use of image resolution to balance accuracy and resource constraints. The proposed system introduces three key components: LLM-based scale inference, knowledge distillation, model disagreement-based sampling. The proposed approach has been demonstrated to improve recognition accuracy while reducing the need for HR imagery.

**Strengths:**

This paper focuses on a highly interesting problem with significant industrial application potential.

The proposed knowledge distillation method and model disagreement-based sampling strategy are both innovative and effective. The distillation enhances the performance of LR models, while the sampling strategy efficiently balances the trade-off between cost and accuracy.

**Weaknesses:**

The approach heavily relies on the examples provided in the prompt for the LLM. The model’s understanding of the relationship between manually defined categories and image resolutions is largely based on these examples. This dependence raises concerns about the generalizability of the method to niche or less common satellite imagery concepts.

The paper does not include experiments comparing the proposed method with techniques that generate HR images via super-resolution, then combine these images with LR data for recognition. Such comparisons, focusing on both computational efficiency and accuracy, would provide a more comprehensive evaluation of the proposed approach.

**Questions:**

The paper distills a model from paired HR and LR images to predict whether HR images are necessary based on prediction differences. Why not directly train a super-resolution model on paired HR-LR images, use it to generate HR images for any LR input, and then perform recognition on both the generated HR and original LR images? If the recognition results differ significantly, HR images could be invoked. What are the advantages of the proposed, more complex method over this approach?

Impact of resolution ratio on the proposed method is suggested to be discussed. Does the resolution ratio between high and low-resolution images affect the performance of the proposed method? Could you please provide some analysis or experiments to explore how this ratio influences the model’s efficiency and accuracy?

---

> ### Author Response · Authors · 2024-11-17
> **Review Response**
>
> Thank you for your insightful comments and review, we address your questions and concerns below:
>
> Weaknesses:
>
> 1. Yes indeed our approach relies on in-context examples. LLMs have surprisingly broad knowledge on the world and may be able to extrapolate more information than we would expect, but one could provide additional in-context examples or further prompting to allow the LLM to generalize to more niche concepts.
>
> We use an off-the shelf LLM, specifically ChatGPT 3.5, here. The LLM semantically understands the scale of different concepts, for example if one were to ask “how large is a tennis court”, most LLMs are able to answer with high levels of accuracy. With concept scale information, we can also ask the LLM whether or not the concept would be visible in a given resolution of a satellite image.
> A good way to think about it is that if the LLMs can answer “Is a tennis court better suited to a 10m per pixel resolution image or a 1m per pixel resolution image, given that a swimming pool is suited to 1m and a coastline is suited to 10m”.  Then for any concept, regardless of how commonly it is searched for within satellite imagery, the LLM would know roughly its size and then be able to infer its visibility for a given resolution.
>
> An alternative/future work can explore prompting the LLM with the true scale and specific resolutions of the satellite. We will discuss this limitation further in the weaknesses section of our paper.
>
> 2. A lot of the use cases of satellite imagery require accuracy. With any in-painting or super resolution techniques, there is room for the model to hallucinate. Take for example researchers in the fields of anthropology [1], who report to Congress or the United Nations about specific incidents of cultural erasure. For the concepts they are keeping an eye on, there’s essentially 0 tolerance for hallucinations in these cases as this work can have major socio-political implications.
>
> [1] ©2023, C. H. W. (2024b, June 1). Between the wars. ArcGIS StoryMaps. https://storymaps.arcgis.com/stories/e1c69b7dd46f4c839dffc0fab9248368

---

> ### Author Response · Authors · 2024-11-17
> **Response to Questions**
>
> Questions:
>
> 1. The idea of using a super resolution model in place of the KD model for model disagreement based sampling is an insightful suggestion! We test this hypothesis using a SoTA super resolution (SR) model [2] and present the results below.
>
> Generating and performing inference using the SR data proved to be significantly more computationally expensive. Here we report inference time, one can see the process is significantly slower than using the KD model on the LR images. This is due to the fact that the SR model accepts 32x32 pixel patches of Sentinel-2 images at a time. So for our test set of 5015 LR images of size 256x256 pixels, the SR model is performing inference on  320,960 patches. This process took 12,549.90 s (~3.5 hours). Additionally upon stitching these SR patches to recreate the coverage of the original LR image, we must split the stitched SR image into 100 components corresponding to the same organization of HR images. This leads to inference using the HR model on 501,500 images. And even after this, one would still need to run the KD model for regions not covered by the budget. Overall this significant jump in inference time makes using any superresolution model in place of our KD model less favorable.
>
> Additionally, our KD model is specifically trained to identify cases where HR data would provide meaningful improvements in recognition accuracy. In contrast, SR models attempt to reconstruct all high-frequency details, regardless of their importance to the recognition task. This targeted approach makes our method more efficient at identifying truly necessary cases for HR image use.
>
> | Model | Seen mAP || Unseen mAP || HR images in Mill. | runtime in s |
> |:-------:|:---:|:---:|:---:|:---:|:---------------------:|:--------------:|
> | | @20 | @100 | @20 | @100 | | |
> | GRAFT HR | 0.501 | 0.513 | 0.541 | 0.574 | 25,163 | 1,539 |
> | GRAFT LR + SR + LLM| 0.527 | 0.583 | 0.512 | 0.547 | **5,954**| 12,867 |
> | GRAFT **KD** + SR + LLM | 0.554 | 0.610 | **0.556** | **0.589**| **5,954** | 12,884 |
> | GRAFT (Ours Full) | **0.633** | **0.639** | 0.502 | 0.564 | **5,954** | **372** |
>
> Here we present the results using the SR images with the HR model to calculate the model disagreement score to sample HR imagery.  LR + SR + LLM denotes the use of both the LLM and model disagreement components of our work, (using the SR model disagreement), without the use of any KD model. KD + SR + LLM denotes results with our full system, but with the model disagreement scores being obtained via the SR imagery.
>
> While it does offer improvements in mAP for unseen classes, the inference time alone is 34 times longer than our technique, and this improvement only comes with the use of our KD model.
>
> 2. The impact of resolution ratio is an important one, we discuss this limitation on line 513.
> Intuitively, if the ratio is closer to 1, then one would expect there are fewer concepts that the models would disagree on as there will be considerable overlap between which concepts have visual signatures and vice versa as the ratio approaches 0.
> However in practice we found that in cases where the resolution ratio was 1:100 (as is the case for our NAIP benchmark) or  1:4 (as is the case in our NICFI benchmark), both KD model disagreement scores were highly correlated with their HR counterparts.
>
> With the correlation coefficient between the disagreement scores of the HR (NAIP) vs KD with the LR model is 0.9322, which signifies a strong positive correlation and the correlation coefficient between the disagreement scores of the HR (NICFI) vs KD with the LR model is 0.998, which also signifies a strong positive correlation. To carry out this experiment we calculate the ranked Spearman correlation coefficient between the two sets of disagreement scores.
>
> [2] Wolters, Piper, Favyen Bastani, and Aniruddha Kembhavi. "Zooming out on zooming in: Advancing super-resolution for remote sensing." arXiv preprint arXiv:2311.18082 (2023).

---

> > ### Author Response · Authors · 2024-12-02
> > **We hope we have addressed your concerns**
> >
> > We hope we have addressed your concerns and have updated the pdf with results and clarifications from the review discussion. Are there any further clarifications needed?

---

### Official Review · Reviewer_JoCu · 2024-11-03

**Soundness:** 3
**Presentation:** 3
**Contribution:** 3
**Rating:** 6
**Confidence:** 4

**Summary:**

This paper introduced a new approach to scale-aware recognition in satellite imagery under resource constraints. The proposed system leverages LLMs and knowledge distillation technique. The proposed system outperformed the baselines  more than 26% mAP in zero-shot techniques, and more than 8% mAP in supervised techniques using 5× fewer HR images.

**Strengths:**

1. This paper presents a novel framework to perform the tradeoff between cost and accuracy.
2. The topic is interesting and valuable.
3. The writing is clear.
4. It is good that the authors plan to release their data and download scripts.

**Weaknesses:**

1. It is unclear how the system handles noisy labels using OpenStreetMap annotations.
2. Can the proposed system retrieve multiple targets simultaneously?
3. How to deal with the time discrepancy between high- and low-resolution images, which can lead to disagreements between the LR and HR models. Did the authors consider the impact of this issue?
4. No mention of runtime in Table 3.

**Questions:**

The system is based on many ideal assumptions. Whether deployed online or offline, it will encounter foreseeable issues such as data storage and retrieval, online search, and download stability. Additionally, the overall process time and reliability are concerns. Although I like the idea, there seems to be a large gap between the proposed method and practical application.

---

> ### Author Response · Authors · 2024-11-17
> **Review Response**
>
> Thank you for your review and questions. We hope to address your concerns and answer your questions below:
>
> Weaknesses:
> 1. We discuss the limitations of OpenStreetMaps in the limitations subsection of our Results (Line 518).  If there are any other datasets the reviewer recommends we use to demonstrate the accuracy of our models, we are happy to do so as we believe it would strengthen our results.
>
> As reported in the paper, we acknowledged that OpenStreetMaps data contains noise, due to its crowd-sourced nature. Unfortunately, there are not many sources for fine-grained labeling of satellite imagery. Therefore, we follow past work in using OSM for training and evaluation (Mall et al., 2024; Bastani et al., 2023). Moreover other standard sources of labeled satellite imagery such as Microsoft’s building footprint dataset work with similar noise levels (Bing Maps, 2018). Additionally, to show robustness to this noise, we performed an experiment using the GRAFT LR model, wherein we constructed bootstrap samples by sub-sampling within our original test set 1000 times. The results show that the 90% confidence interval is within 2.43% of the reported mAP.
>
> 2. Yes we can retrieve multiple targets simultaneously. Each model provides the scores for all query concepts at once as logits. So for all images, we run inference using the LR model just once. If some of the query concepts require HR imagery, we run inference on the images with the KD model (again just once). We then calculate the disagreement (which is shared across all concepts), and retrieve how many ever HR images the budget allows for and evaluate using the HR model only once.
>
> 3. We address time discrepancies on line 324 in the Benchmark subsection. “To handle time discrepancies we retrieved the LR image for a set of HR within the same month that the HR image was collected to avoid major disagreements between the two sets of images.”
>
> 4. We do not include runtime in tables 3 and 4 as these tables are used to demonstrate ablation results on LR imagery alone. We include runtimes for **all** of the techniques in tables 1 and 2 (page 8).
>
> Questions:
>
> Deployment: Indeed, one would need to build appropriate storage systems for practical deployment. But storage systems for data  and detection models like this are already being done manually by anthropologists, archaeologists, soil and crop scientists, amongst many others. We offer an efficient alternative to the manual overhead, and we believe there is limited additional cost over how retrieval in satellite imagery is currently being done. Our focus is on tackling retrieval and diving deeper into how the size and scale of a concept is best suited to different resolutions, and how this information can be used to achieve better retrieval results given cost constraints.
>
> We further believe that the systems challenge notwithstanding, the problem of effective retrieval in the face of the cost of data is a pervasive challenge in many applications. Many researchers in the fields of anthropology [1], among others, manually search through large amounts of HR satellite imagery to pinpoint sites of interest and then purchase HR imagery to monitor these sites of interest a few times a year (2-12) due to high costs. Our technique would greatly help in both the initial search for points of interest and for monitoring for these sites in between the HR image acquisitions, and the storage space and retrieval needs of the LR imagery is far lower than that of the HR imagery. We hope that including the information regarding the scale of the datasets and the inference runtime help alleviate concerns regarding overall process time and deployment.
>
> [1] ©2023, C. H. W. (2024b, June 1). Between the wars. ArcGIS StoryMaps. https://storymaps.arcgis.com/stories/e1c69b7dd46f4c839dffc0fab9248368
>
> [2] Mall, Utkarsh, et al. "Remote Sensing Vision-Language Foundation Models without Annotations via Ground Remote Alignment." ICLR 2024
>
> [3] Bastani, Favyen, et al. "Satlaspretrain: A large-scale dataset for remote sensing image understanding." ICCV 2021
>
> [4] Microsoft. (2018). Microsoft/USBUILDINGFOOTPRINTS: Computer generated building footprints for the United States. GitHub. https://github.com/microsoft/USBuildingFootprints

---

> > ### Author Response · Authors · 2024-12-02
> > **We hope we have addressed your concerns**
> >
> > We hope we have addressed your concerns and have updated the pdf with results and clarifications from the review discussion. Are there any further clarifications needed?

---

### Official Review · Reviewer_nAgQ · 2024-11-04

**Soundness:** 3
**Presentation:** 3
**Contribution:** 4
**Rating:** 8
**Confidence:** 4

**Summary:**

To provide a holistic treatment of scale to achieve accurate recognition under a budget, the paper tries to give solutions to which resolution is the best and where and when  the costlier higher-resolution (HR)imagery should be acquired. The main contributions are: (1)knowledge distillation is adopted to allow Low resolution models to improve significantly in  recognizing finer concepts by learning from High resolution ones. (2) Semantic understanding of LLM is used to determine the  scale of each concept.(3) A sampling strategy for HR imagery based on model disagreement is proposed. The process is frameworked as remote sensing image retrieval task. Experiments show some promising success for the proposed idea.
The idea is very novel and interesting.

**Strengths:**

1. A novel idea of scale determination is proposed for targer recognition using reasonable resolution remote sensing images.
2. Low resolution detection model using distillation strategy is taken as the replacement of high resolution detection model.
3. Two benchmark datasets are constructed for research.

**Weaknesses:**

1. The budget constraints seems actually impose no impact on the final result.
2. The processes such as distillation, concept scale LLM training are not provided the details.
3. The retrieval process has no systematic description to include all the subprocesses.

**Questions:**

1.Quote" A location l with higher δ(l) is more likely to have concepts that MHR could detect where MLR  could not (or vice versa)." Since it is the absolute value, it is difficult to understand.
2. How is the LLM trained? What's the confidence for the determination of LLM?
3. Can it be understood as that the low resolution images taken as large as possible because the cost constraints is not imposed?
4. Quote"...Remote CLIP(Liuetal.,2024) in Table 3." But the table is not the detection result.

---

> ### Author Response · Authors · 2024-11-17
> **Review Response**
>
> Thank you for your insightful review and feedback. We hope to address your concerns and answer your questions below:
>
>
> Weaknesses:
>
> 1. The final result is impacted by budget, in the sense that we are able to have improved retrieval results with far fewer HR images. We can look to Table 2, our model (bottom row) outperforms the solely HR model. We are able to get an improved accuracy on a tight budget since we only sample HR images when necessary. The reason we are able to do so is (a) some concepts are better in LR and (b) for the remaining concepts we are able to recognize which regions require HR. Of course if the budget decreases further, at some point, the accuracy will also decrease. We demonstrate this in the table below with results using our method for varying budgets.
>
>
>
> | Model | Seen mAP || Unseen mAP || HR images in Mill. | runtime in s |
> |-------|:---:|:---:|:---:|:---:|:--------------------------------------:|:-------------:|
> | | @20 | @100 | @20 | @100 | | |
> | GRAFT HR | 0.501 | 0.513 | **0.541** | **0.574** | 25,163 | 1539 |
> | GRAFT LR | 0.482 | 0.507 | 0.379 | 0.471 | 0 | 17|
> | GRAFT (Ours Full), Budget 100 | 0.557 | 0.605 | 0.439 | 0.521 | **595** | 48|
> | GRAFT (Ours Full), Budget 500 | 0.601 | 0.626 | 0.480 | 0.542 | 2,977 | 170 |
> | GRAFT (Ours Full), Budget 750 | 0.614 | 0.617 | 0.495 | 0.557 | 4,466 | 247 |
> | GRAFT (Ours Full), Budget 1000  | **0.633** | **0.639** | 0.502 | 0.564 | 5,954 | 372 |
>
> 2. We will add further details on each of these components and release the code upon acceptance. If there are any specific clarifications we are happy to address them! With regards to the LLM, we use an off-the shelf LLM specifically ChatGPT 3.5 which is free to the public and we use the prompts specified in the supplemental materials.
>
> 3. We are happy to clarify the specifics of our retrieval process, as described in section 3.6 (lines 272-281). Given a concept, our LLM determines which resolution is best suited. If the best suited resolution is LR, we perform inference using the LR model and all LR imagery only. If the best suited resolution is HR, we ask the user for their HR budget. Then we perform model disagreement using LR imagery alone and the LR and KD models. With these disagreement scores and the user budget, we then sample the top locations, obtain HR images and perform inference using the HR model for these locations. The retrieval results for the remaining area, that it is preferable to use HR for, but cannot be fit in the budget,  is scored using the KD model on LR imagery.
>
> Questions:
>
> 1. Our model disagreement component only comes into play when we are searching for concepts we know are better detected in HR. So for these concepts, if there is any disagreement between the HR and LR models, regardless of the direction of the disagreement, we know to sample HR imagery.
>
> 2. As we mentioned earlier, we use an off-the shelf LLM specifically ChatGPT 3.5, and provide in-context examples as described in section 6.3 (in the supplementary material). Here we compare our LLM approach to other baselines, and we will include additional results using other off the shelf LLMs (Gemini and Claude). We are happy to provide more details if you have any further questions.
>
> | Technique | Seen Classes Accuracy | Unseen Classes Accuracy |
> |:-----------:|:---------------------:|:----------------------:|
> | LLM ChatGPT 3.5 (ours) | **100%** | **100%** |
> | LLM Gemini (ours) | **100%** | **100%** |
> | LLM Claude (ours) | **100%** | **100%** |
> | OSM | 73.33% | 90% |
> | Always HR | 83.33% | 90% |
> | Always LR | 16.67% | 10% |
>
> We compared our LLM-based approach for inferring the best modality per concept with other baseline approaches. For each approach, we evaluate its accuracy in terms of determining the right modality. These baselines include either always choosing the HR modality and always choosing LR modality, or selecting between the two based on the average area of each concept, as provided via OpenStreetMaps (OSM). Here one can see that the LLM performs with 100% accuracy for our set of concepts, but it is also capable of easily being extended to far more concepts.
>
> 3. In some senses, yes, we can use the largest possible LR image, however in practice we fix the image size and use more tiles.
>
>
> 4. You are right! Thank you for pointing this out to us, this section heading is a typo. We apologize for any confusion and will correct it to “retrieval” in the revision.

---

> > ### Author Response · Authors · 2024-12-02
> > **We hope we have addressed your concerns**
> >
> > We hope we have addressed your concerns and have updated the pdf with results and clarifications from the review discussion. Are there any further clarifications needed?

---

### Author Response · Authors · 2024-11-17
**Response to Reviewers**

We thank the reviewers for their time and feedback. Here we answer common questions, we address individual concerns in the comments.

1. **Overall Retrieval Process**  Reviewers nAgQ, JoCu, and Cazh had questions about details of our retrieval process. Given a concept, our LLM determines which resolution is best suited. If the best suited resolution is LR, we perform inference using the LR model and all LR imagery only. If the best suited resolution is HR, we ask the user for their HR budget. Then we perform model disagreement using LR imagery alone and the LR and KD models. With these disagreement scores and the user budget, we then sample the top locations, obtain HR images and perform inference using the HR model for these locations. The retrieval results for the remaining area, that it is preferable to use HR for, but cannot be fit in the budget,  is scored using the KD model on LR imagery.

2. **LLM Component** Reviewers nAgQ, Kyd3, and CazH had questions about the LLM component of our technique. We use an off-the shelf LLM, specifically ChatGPT 3.5, and provide in-context examples as described in section 6.3 (in the supplementary material). We discuss testing the LLM on **unseen** classes in both sec. 4.2.4 on page 9 as well as more in depth on page 2 of the supplementary materials. Here we compare our LLM approach to other baselines, and we include additional results using other off the shelf LLMs (Gemini and Claude).

| Technique | Seen Classes Accuracy | Unseen Classes Accuracy |
|:-----------:|:---------------------:|:----------------------:|
| LLM ChatGPT 3.5 (ours) | **100%** | **100%** |
| LLM Gemini (ours) | **100%** | **100%** |
| LLM Claude (ours) | **100%** | **100%** |
| OSM | 73.33% | 90% |
| Always HR | 83.33% | 90% |
| Always LR | 16.67% | 10% |


We compared our LLM-based approach for inferring the best modality per concept with other baseline approaches. For each approach, we evaluate its accuracy in terms of determining the right modality. These baselines include either always choosing the HR modality and always choosing LR modality, or selecting between the two based on the average area of each concept, as provided via OpenStreetMaps (OSM). Here one can see that the LLM performs with 100% accuracy for our set of concepts, but it is also capable of easily being extended to far more concepts.

3. **Impact of Different Budgets** Reviewer nAgQ inquired about the impact of various budgets on performance. We are able to get an improved accuracy on a tight budget since we only sample HR images when necessary. The reason we are able to do so is (a) some concepts are better in LR and (b) for the remaining concepts we are able to recognize which regions require HR.

We demonstrate this in the table below with results using our method with varying budgets.
| Model | Seen mAP || Unseen mAP || HR images in Mill. | runtime in s |
|-------|:---:|:---:|:---:|:---:|:--------------------------------------:|:-------------:|
| | @20 | @100 | @20 | @100 | | |
| GRAFT HR | 0.501 | 0.513 | **0.541** | **0.574** | 25,163 | 1539 |
| GRAFT LR | 0.482 | 0.507 | 0.379 | 0.471 | 0 | 17 |
| GRAFT (Ours Full), Budget 100 | 0.557 | 0.605 | 0.439 | 0.521 | **595** | 48|
| GRAFT (Ours Full), Budget 500 | 0.601 | 0.626 | 0.480 | 0.542 | 2,977 | 170 |
| GRAFT (Ours Full), Budget 750 | 0.614 | 0.617 | 0.495 | 0.557 | 4,466 | 247 |
| GRAFT (Ours Full), Budget 1000  | **0.633** | **0.639** | 0.502 | 0.564 | 5,954 | 372 |

5. **Use of Super Resolution model for Model Disagreement Score** Reviewer Kyd3 suggested using a super resolution (SR) model in place of the KD model for model disagreement.

Here we present the results using the SR images with the HR model to calculate the model disagreement score to sample HR imagery. LR + SR + LLM denotes the use of both the LLM and model disagreement components of our work, (using the SR model disagreement), without the use of any KD model. KD + SR + LLM denotes results with our full system, but with the model disagreement scores being obtained via the SR imagery.

With the use of our KD model, SR does offer improvements in mAP for unseen classes. However, the total inference time alone is ~**34 times longer than our technique**, and **this improvement only comes with the use of our KD model.**  Overall this significant jump in inference time makes using any superresolution model in place of our KD model less favorable.

| Model | Seen mAP || Unseen mAP || HR images in Mill. | runtime in s |
|:-------:|:---:|:---:|:---:|:---:|:---------------------:|:--------------:|
| | @20 | @100 | @20 | @100 | | |
| GRAFT HR | 0.501 | 0.513 | 0.541 | 0.574 | 25,163 | 1,539 |
| GRAFT LR + SR + LLM| 0.527 | 0.583 | 0.512 | 0.547 | **5,954**| 12,867 |
| GRAFT KD + SR + LLM | 0.554 | 0.610 | **0.556** | **0.589** | **5,954** | 12,884 |
| GRAFT (Ours Full) | **0.633** | **0.639** | 0.502 | 0.564 | **5,954** | 372 |

Thank you all for your time and we are happy to answer any further questions.

---

### Meta-Review · Area_Chair_yACF · 2024-12-23

**Metareview:**

The paper proposes a method for scale-aware scene recognition in satellite imagery. This method utilizes knowledge distillation to transfer knowledge from models trained on high-resolution images to models trained on low-resolution images. The goal is to enable the latter to work alongside low-resolution models to identify regions that might require high-resolution imagery based on low-resolution images only. Model disagreements are employed as a decision-making mechanism to determine whether high-resolution imagery is necessary for a given geographical region based solely on low-resolution images.

In this context, it is debatable whether the use of large language models (LLMs) is necessary to identify the scale of a concept. Model disagreement alone can serve as an effective decision mechanism, making scale information from LLMs redundant. While the reviewers recognize the overall idea as interesting and acknowledge the value of model disagreement in determining whether cost-intensive high-resolution imagery is needed, it is questionable whether this approach constitutes a novel contribution.

**Additional Comments On Reviewer Discussion:**

The authors adequately addressed the reviewers' comments during the rebuttal phase, which led to a score improvement from CazH.

---

### Decision · Program_Chairs · 2025-01-22

Accept (Poster)